# Bioactive Compounds for Customized Brain Health: What Are We and Where Should We Be Heading?

**DOI:** 10.3390/ijerph20156518

**Published:** 2023-08-03

**Authors:** Lina Begdache, Rani Marhaba

**Affiliations:** 1Health and Wellness Studies Department, Binghamton University, Binghamton, NY 13902, USA; 2Norton College of Medicine, SUNY Upstate Medical University, Syracuse, NY 13210, USA

**Keywords:** bioactive compounds, customized diet, gender differences, Mediterranean diet, nuclear receptors

## Abstract

Many strides have been made in the field of nutrition that are making it an attractive field not only to nutrition professionals but also to healthcare practitioners. Thanks to the emergence of molecular nutrition, there is a better appreciation of how the diet modulates health at the cellular and molecular levels. More importantly, the advancements in brain imaging have produced a greater appreciation of the impact of diet on brain health. To date, our understanding of the effect of nutrients on brain health goes beyond the action of vitamins and minerals and dives into the intracellular, molecular, and epigenetic effects of nutrients. Bioactive compounds (BCs) in food are gaining a lot of attention due to their ability to modulate gene expression. In addition, bioactive compounds activate some nuclear receptors that are the target of many pharmaceuticals. With the emergence of personalized medicine, gaining an understanding of the biologically active compounds may help with the customization of therapies. This review explores the prominent BCs that can impact cognitive functions and mental health to deliver a potentially prophylactic framework for practitioners. Another purpose is to identify potential gaps in the literature to suggest new research agendas for scientists.

## 1. Introduction

It is well-recognized that regular consumption of a nutrient-dense diet is advantageous for physical health. With the recent advancements in technology and imaging studies, it is becoming more evident that a nutrient-dense diet has a significant impact on brain health as well. The translation of high-resolution studies is documented in epidemiological reports that link low-nutrient-dense diets, such as the Western diet (WD), to mental health ailments and high-quality diets, such as the Mediterranean diet (MD), to mental well-being [1,2,3]. A nutrient-dense diet (ND) typically encompasses a spectrum of naturally colorful and whole food such as legumes, fruits, vegetables, nuts, herbs, spices, and grains. Several constituents of these diets promote health at the cellular level through the formation of metabolites, improvement of the gut microbiota profile, activation of cellular signaling, and modulation of gene expression. The beneficial components of ND are typically phytochemicals, except for a few that are commonly known as bioactive compounds (BCs). They confer several health benefits due to their large repertoire of cellular activities and, hence, the nomenclature bioactive. They are signaling molecules that modulate the process of gene expression to impact the quantity and quality of protein production. In addition, BCs hold several therapeutic qualities ranging from antidiabetic and anti-cancerous to antidiuretic and anti-atherosclerotic properties [4], which make them attractive to the field of medicine. Their neuroprotective role is slowly emerging, which may hold a prophylactic defense against neurodegenerative diseases [5]. However, due to the limited nutrition education in the medical curriculum, doctors are apprehensive about applying nutrition therapies to patient care.

Imaging and epidemiological studies suggest that there is a difference in brain morphology between men and women that explains several of their behavioral and health outcomes [6,7,8,9]. This new evolving concept suggests that men and women should be studied independently of each other to provide an accurate conclusion.

Therefore, the purpose of this review is to explore the prominent bioactive compounds regarding cognitive functions and mental well-being to deliver a potentially prophylactic framework for practitioners in the hope to revamp the need to include a didactic nutrition curriculum in medical education. A second purpose is to identify potential gaps in the literature to suggest new research agendas to possibly encourage a call to action for agencies to develop more robust funding streams for these gaps.

### The Evolution of the Nutritional Field

The connection between nutrition and health dates back to Hippocrates, the father of modern medicine, and his followers, who described several disease conditions [10] and eloquently addressed the impact of diet on health. While great strides were made in the field of nutrition, a longstanding practice was to describe the relationship between diet and health, mostly on physiological and biochemical bases. The health benefits of ND were initially ascribed to the action of vitamins and minerals. In the early twentieth century, the work of geneticists, structural chemists, and physicists led to the emergence of the molecular biology field [11]. Fast forward, these disciplines directed the birth of the genomic era, thanks to the rapid evolution of molecular biology techniques. Today, the newly born field of molecular nutrition describes the impact of nutrition on health through intracellular signaling, nutritional metabolites, and nuclear receptors that collectively influence gene expression. These discoveries, coupled with enhanced brain imaging and molecular techniques, branched molecular nutrition into the nutritional neuroscience and nutritional psychiatry fields. In fact, the diet is emerging as a potential first line of defense against cognitive decline and mental health ailments [8,12,13]. These research fields are strengthening the therapeutic role of diet and extending the role of dietitians and nutrition scientists to cover an expanding horizon that holds multilevel promises.

Therefore, reforms in the didactic dietetic and nutrition programs are needed to meet the fast-paced progression of the field. The incorporation of molecular nutrition education in medical school curricula is essential to help future medical doctors appreciate the prophylactic and therapeutic effects of nutrients.

## 2. The Human Brain

The brain is a complex high-maintenance organ with established barriers and limited antioxidative capacity that make it susceptible to biochemical insults. Although it is only 2% of body weight, it is an expensive organ that consumes about 20% of daily energy [14]. Neurons are terminally differentiated cells, which means that the loss of these brain cells is irreplaceable by cell division of existing neurons. Although neurogenesis is an auxiliary option, it is highly regulated by a myriad of restraining conditions. Fortunately, the brain evolved the blood–brain barrier (BBB), a tight meshwork of endothelial cells that control molecular exchange to regulate the internal and external environment of neuronal tissue and its surrounding fluid milieu. Therefore, the BBB plays a vital role in providing the necessary nourishing and protective chemicals to the brain [15]. Unfortunately, the BBB is susceptible to anatomical damage, which renders it a leaky junction and an injurious intersection. Within the brain, reactive oxygen species (ROS) and reactive nitrogen species (RNS) are highly potent by-products of enzymatic and non-enzymatic reactions that could induce further damage. In fact, neurodegeneration and loss of brain volume have been associated with excessive production of ROS/RNS. The reactivity of these molecules with proteins, lipids, and DNA creates an internal mayhem that culminates in apoptosis and/or necrosis of brain cells, leading to neuroinflammation. When considering the diet in relation to brain health, the research has eloquently shown that a nutrient-dense diet, such as MD, satisfies the affluent requirements of the brain and potentially protects the BBB from injury and inflammatory processes [16]. Such diets have been linked to mental well-being and higher brain volumes with age [17,18].

### 2.1. Should the Human Brain Be Better Known as the Gender-Based Brain?

There is enough evidence to support the differential impact of diet on brain function and health of men and women [6,19,20]. Although anatomically common characteristics exist between men and women’s brains, mounting evidence suggests that their brains are morphologically different [6,20]. The major variances reported are the degree of connectivity between diverse brain regions, type of circuitry activation, and brain region recruitment under certain tasks [21]. These differences suggest that a differential repertoire of nutrients may be needed to support brain health and functionality in men and women. Although studies on the differential effects of BC metabolism among men and women are scarce, they are starting to emerge [22]. Studying the impact of BCs on the gender-based brain may potentially lead to some interesting findings. The diet has the potential to impact mental health with a resilience effect, potentially through epigenetic modifications [2]. Consequently, failure to meet the nutritional demands of the brain may alter the epigenome and increase the risk of mental health issues with a risk of recurrence. In addition, females have twice the risk of mental disease than males [23]. Although the depressed mood of women tends to be associated with reproductive events such as pre-menses, postpartum, and peri-menopause due to the fluctuation of their reproductive hormones [24], some research suggests the need for a customized diet to improve mental well-being in women. Women are more likely to require a spectrum of nutrient-dense food to achieve mental well-being than men [19], presumably due to the denser connectivity between brain regions [7]. In addition, women’s mood appears to be more sensitive to high-glycemic index food and caffeine when compared to men [25]. This could be explained by the different endocrine physiological processes between men and women. Therefore, the differential impact of the diet on biological sex differences is an area that could be further explored.

### 2.2. Sex Hormones and Brain Health

Gonadal hormones exert their function through their corresponding nuclear receptors. Several genes involved in brain homeostasis are regulated by sex hormones. Estrogen controls gene expression by binding to the estrogen receptors (ERs), which activates transcriptional processes and/or signaling cascades. Likewise, androgens exert their genomic and non-genomic effects though the androgen receptors (ARs). These are highly regulated complex mechanisms. Several of these upregulated genes are linked to the metabolism of nutrients. For instance, the phosphatidylethanolamine-N-methyltransferase (PEMT) gene is activated by estrogen and catalyzes the de novo synthesis of choline (from phosphatidylcholine), which is crucial for the production of the neurotransmitter acetylcholine and regulation of the dopaminergic system [26]. Aberrant PEMT activity was linked to neurological dysfunction [27,28]. Men and menopausal women have higher nutrient needs for choline [29], which directly impacts their brain health, supporting the notion of personalized dietary therapy.

### 2.3. Bioactive Compounds: A Hidden Potential Brain Remedy?

Several BCs have been described to confer neuroprotective, neuroplastic, neurogenesis, and anti-inflammatory properties [30,31,32]. However, most studies have focused on one compound or food group at a time and mostly in regard to neurodegenerative diseases [5,33]. However, few reports were published on the effect of BCs on brain health to describe cognitive functions and mental health [34]. Due to the limitations in studying these characteristics in humans, most evidence comes from in vitro and laboratory animal models that describe their cellular and molecular mechanisms. However, their defined effects have been reported in epidemiological studies that broadly suggest that these bioactive compounds may provide similar protective properties against neuroinflammation and neuro-oxidation, although appropriate dosage and proper mechanisms have not been elucidated in men and women. In fact, several neurological conditions benefit from dietary adjustments [35]. This suggests that a nutrient-dense diet, potentially high in BCs, may protect the brain from several ailments. Since neurons are terminally differentiated cells, it is crucial to act prophylactically rather than therapeutically.

Substantial work has been achieved by the scientific community in deciphering the role of BCs on brain health and function. Adherence to the Mediterranean diet (MD) and the Mediterranean-DASH Intervention for Neurodegenerative Delay (MIND) diet has been associated with a slow deterioration in cognitive abilities in seniors when compared to poor-nutrient diets such as the Western diet (WD) [18,36,37,38]. The latter is typically composed of fast food, meat, and starchy carbohydrates lacking a myriad of phytochemicals. Deficiencies in these active compounds have been linked to mental distress and brain atrophy [1]. Although aging is a risk factor for brain atrophy as oxidative stress increases with age [39], WD appears to accelerate this process [40]. On the other hand, MD has been linked to larger brain volume with age and mental well-being [17,18]. MD is high in olive oil, fruits, nuts, abundant use of herbs and spices, vegetables, whole grain, legumes, and fish and tends to be moderate in red meat and sweets. Most studies combined findings from both men and women and reported an association between diet quality, such as a plant-forward diet, and mental functions. However, studies that investigated the diet based on gender reported differential effects [13,19,25,41]. For instance, a plant-based diet may be more beneficial to women’s mood than men’s.

BCs support brain health as evidenced by their association with a lower risk of dementia, better cognition, as well as executive functions with aging [42,43]. Although the studies on BCs and brain health are still in their infancy stage, the proposed mechanisms of action comprise direct and indirect effects. The direct effect includes free radical scavenging, improvement in the cellular antioxidative machinery, regulation of apoptosis, and improvement in neurotransmission [5,44]. The indirect properties consist of improvement in cerebral vascular flow and enhancement of the gut microflora profile [45]. The most prominent BCs studied and described in the literature are phenolic compounds, phytoestrogen, resveratrol, carotenoids, organosulfur compounds, isothiocyanates, and omega-3 fatty acids, among others. However, their functions in the brain are slowly emerging. In fact, the consumption of a plant-forward diet has been associated with a lower incidence of chronic diseases [46]. BCs have been extensively studied as anti-cancer agents with apoptotic and cytotoxic effects [47]. Their health benefits on the cardiovascular, gastrointestinal, integumentary, and immune systems have been elucidated as well [48,49,50,51]. Indeed, phytochemical-rich foods, especially the ones high in flavonoids, were reported to be inversely associated with age-related cognitive deficits in animals and human models [52,53,54]. However, studies assessing the role of BCs in mental health are still limited.

### 2.4. Neuroprotective Signaling of Bioactive Compounds

The neuroprotective mechanisms of BCs are mainly through the activation of several intracellular signaling pathways. The growth and survival mitogen-activated protein kinase (MAPK) pathway is crucial for neuronal survival, and differentiation. The PI3K/AKT (Phosphoinositide 3-kinases/Akt) pathway averts apoptosis by promoting the expression of Bcl-2, the Nrf2/ARE regulates redox signaling and cellular homeostasis, and protein kinase C modulates the cell regulatory pathway [55]. The Akt-ERK1/2 pathway phosphorylates Bad (BCL2-associated agonist of cell death) and Bim (Bcl-2-interacting mediator) to prevent apoptosis [56]. Although most BCs epigenetic modification studies focused on cancer and the reversal of aberrant tags, there is some level of understanding of how these compounds modulate the epigenome. BC components accomplish their role with great precision on at least two levels of gene expression, as chemical regulators of epigenetic mechanisms and as factors directly controlling the activity of nuclear receptors. In fact, several BCs modify the chromatic structure to modulate gene expression by directly inhibiting DNA methyltransferases or histone-modifying enzymes or by altering the availability of substrates necessary for those enzymatic reactions [57]. There is evidence that those who adhere to ND develop mental well-being with resilience [2], which suggests a potential epigenetic role for BCs and other nutrient-dense food in brain neuroplasticity.

### 2.5. Nuclear Receptors

Several BCs directly affect gene expression by serving as ligands for nuclear receptors (NRs). The latter are a family of transcription factors that act as sensory receptors to monitor the external environment though the intracellular messaging system [58]. NRs are activated by a myriad of endogenous hydrophobic ligands ranging from steroids and thyroid hormone to bile acids and dietary metabolites such as xenobiotic lipids [59]. Their activation governs the expression of genes involved in broad biological functions, including metabolism and cellular adaptations to stimuli. The brain hosts many NRs that confer several metabolic and neuroprotective properties. BCs as NR ligands include flavonoids (Daidzein, Genistein, Naringenin, EGCG agonists for several NR) [60], phytosterols (such as sterols and stanols that are Farnesoid X Receptor antagonist) [61], omega-3 fatty acids (peroxisome proliferator-activated receptor agonist) [33], and vitamin D (Vitamin D receptor agonist and not typically listed as BCs) [62] and, therefore, support brain homeostasis through concomitant modulation of several gene activities. In fact, these functions are collectively regulated by a cross-talk between different families of nuclear receptors, such as PPAR and VDR, in establishing brain homeostasis [63,64]. Therefore, the commonly known “healthy diet” provides several bioactive compounds that contribute to multi-level intracellular communication. This internal communication among NRs is crucial for setting homeostatic and adaptive mechanisms.

Several pharmaceuticals mimic the action of dietary BCs [65,66,67] either through cellular signaling or via nuclear receptor activation, some of which hold a nutrient-sensing function. In fact, several BCs are ligands for numerous of these nuclear receptors [68]. However, many of these drugs bear several side effects that are not detected with BC consumption. It is plausible that selective activation of the NRs by certain drugs disturbs the kinetics of NRs and, hence, the internal equilibrium of the cross-talk. Although this notion needs further research for validation, epidemiological studies comparing the impact of ND on health support this idea. If this theory holds true, dietary adjustments to restore this balance through modulation of NR activity may reduce the severity of the side effects. Another potential breakthrough, if proven factual, could be that personalization of diet therapy may reduce or eliminate the need for pharmaceutical use. This potential advancement in the nutrition field will attest to Hippocrates’ statement: let thy food be thy medicine, and thy medicine be thy food. This concept suggests that a proper repertoire of BCs may have a beneficial effect potentially of similar magnitude to these pharmaceuticals if consumed at a therapeutical dosage. With the emergence of the personalized medicine field, there is a need to identify the proper and desirable dosage based on specific health conditions, genotype, gender, age, gut microbiota, and others. Polymorphism of genes and the gut microbial profile can modify their function and physiological responses [45,69]. Therefore, many research opportunities exist in this realm.

## 3. Brain Health and Behavior: Is It Human or Bacterial Control?

The human genome, consisting of 26,000 functioning genes, is among the smaller pool compared to many simpler organisms. The marriage of the Human Genome Project and the Human Microbiome Project brought to light the significant impact of microbial genes on the physiological and behavioral complexity of the human being. Most of the work performed to decipher the role of the human microbial genome was performed on germ-free (GF) laboratory animals. Evidence suggests that microbial metabolites are released into the circulatory system and most often are modified by the host to modulate behavioral and neuroendocrine responses [70]. Some interesting findings related to GF mice, replicated by mice on antibiotics, include lower vigilance and heightened stress with extreme behaviors such as timidity or full-blown aggressiveness [71]. GF animal studies have also shown the implication of the microbiota in several neurological diseases and suggested that fecal transplantation may alleviate some of the symptoms [72,73,74,75]. Over 95% of serotonin is synthesized in the gut. Serotonin is a neurotransmitter that modulates mood and behavior. Its synthesis and bioavailability is managed by the gut microbiota [76,77].

## 4. The Intimate Relationship between the Gut Microbiota and Polyphenols

The human microbiome is composed of a collection of organisms such as bacteria, fungi, viruses, archaea, and protozoans. They colonize different areas of the human body, such as the integumentary system, nasal and oral cavities, gastrointestinal tract, respiratory system, and the female genital tract [78,79]. Over 100 trillion microbial cells colonize the gut, and they contribute different metabolites that impact human health [74]. The composition and the stability of the microbiome differ from one person to another as they are influenced by several internal and external factors. The relationship between polyphenols and the gut microbiota is bidirectional. Polyphenols could act as prebiotics to support the growth of beneficial bacteria and reduce the number of pathogenic ones. Consequently, the beneficial microbiota may process polyphenols to boost their bioavailability. Polyphenols and bacterial metabolites work synergistically to modulate metabolic pathways that support several health outcomes and balance serotonin metabolism and other neurotransmitters [76,80].

### 4.1. The Microbiota–Gut–Brain Communication

The connection between the gut and the brain is bidirectional and mediated by an elaborate network of efferent fibers projecting into the gastrointestinal (GI) tract and afferent fibers that project to several interconnected regions of the brain. The vagus nerve, which is the major fiber mediating parasympathetic autonomic nervous system activity, connects the gut and brain through the gut–brain axis [81]. It senses microbiota metabolites through its afferent fibers and transfers gut information to the central nervous system to generate a response. Therefore, an unhealthy microbiota composition may lead to a maladaptive response in the brain, impacting neurodevelopment and neuropsychology and leading to neurodegenerative diseases [82]. The Western diet contains food devoid of essential nutrients that tend to be high in fat and sugar and low in fiber and BCs. The lack of phytochemicals and the high fat and sugar content alter the gut microbiota, leading to dysbiosis [83]. The by-products of the processed BCs are metabolites that have shown some effective therapeutic effects against depression. However, only a handful of reports to date have depicted mechanistically this symbiotic relationship through the modulation of cellular signaling pathways and neurotransmitter synthesis [84]. The etiology of anxiety and depression are known to be multifactorial. It is described to be a gene–environment interaction in multiple physiological systems [76]. The fact that a therapeutic regime of probiotics and dietary polyphenols improve the symptoms of depression and anxiety points toward the significant impact of the gut microbiota in modulating neurotransmission [85].

### 4.2. Honing in on the Different BCs

#### Phenolic Compounds

Over 8000 naturally occurring phenolic compounds have been reported in the literature [86]. Phenolic compounds are plant chemicals known as phytochemicals that are present abundantly in fruits, vegetables, nuts, olive oil, spices, cocoa, and others. Most of the phytochemicals are produced by plants to confer protection as antioxidant pigments against the UV light-generated free radicals or as repellent to ward off predators [87]. They are also known as polyphenols, which encompass a large group of plant chemicals. They hold several phenolic rings, and their basic chemical structure produces a two-general class of flavonoids and non-flavonoids. Both classes have been described to deliver health benefits due to their wide range of biological activities.

Flavonoids occur as aglycones or as methylated, glycosylated, acetylated, and sulphated products. Collectively, they produce six classes of flavonoids: Flavones, flavanols, flavan-3-ols, flavanones, anthocyanidins, and isoflavones, which are further subdivided into subclasses [88]. Food sources of flavonoids are nuts, fruits, vegetables, and legumes. Non-flavonoids include phenolic acids, stilbenes, and lignans found in purple-red skin fruits, grains, and cereals. Like flavonoids, non-flavonoids were described to have health-promoting properties.

The main dietary groups of flavonoids include: (1) flavones, such as apigenin and luteolin, in cruciferous, parsley, and thyme; (2) flavonols, such as kaempferol, myricetin, isorhamnetin, and quercetin, that are found in onions, apples, teas, and berries; (3) flavan-3-ols, such as epicatechin, epigallocatechin, and epigallocatechin gallate (EGCG), that are in green tea, grapes, and chocolate; (4) anthocyanidins, such as cyanidin, malvidin, and peonidin, that are mostly in berries and eggplant; (5) isoflavones, such as daidzein and genistein, that are in soybeans and soy products.

Water solubility of flavonoids depends on hydrophilic substituents such as sugars and hydroxyl groups, while isopentyl and methyl groups increase the lipophilic characteristics of flavonoids [86].

## 5. Bioavailability and Cellular Mechanisms of Polyphenols

Upon food digestion, released BCs are often chemically modified by intestinal enzymes to improve absorption. It appears that polyphenol diffusion through the BBB is highly dependent on the lipophilic characteristics of the compound. Some evidence suggests that a special repertoire of biochemical modifications is associated with enhanced transport transcellularly through the BBB [89]. The less polar polyphenols and their methylated counterparts tend to have a greater transport affinity for permeation than their polar counterparts [90]. The level of lipophilicity or polarity of the polyphenol dictates its degree of transport by simple diffusion. Those that escape intestinal absorption are further modified by the colonic microbiota into compounds that are readily absorbed [91]. Most often, they are low molecular weight metabolites, which could reach circulation at greater concentrations than their parent chemicals [92]. Low molecular polyphenols, <500 g/mol, appear to cross the BBB through passive permeation [93].

Therefore, the bioavailability of polyphenols in the systemic circulation has been elucidated; however, their capability to competently cross the BBB is still unclear. Their transport to the brain is also dependent on their stereochemistry and degree of interface with efflux transporters at the BBB [94]. In vivo and animal studies suggest that several BCs do cross the BBB and confine in brain tissues, which may explain the epidemiological and experimental studies that propose neuroprotective and neuromodulator effects [94]. As for dosage, the evidence is conflicting. A report stated that BCs cross the BBB at a physiological concentration that represents food intake [90], while some animal and in vitro studies reported similar observations using magnified dosages that do not represent human consumption.

To date, only a handful of BCs have been shown to cross the BBB, which suggests that the claimed associated improvement in brain health is potentially a systemic effect.

### 5.1. Phytoestrogens

Plant-derived phytoestrogens are the main subclass of the polyphenol family. Unlike mammalian estrogen and estradiol, these chemicals are non-steroidal and share a comparable chemical structure with estrogen, which make them a perfect nuclear estrogen receptor agonist (ERs) [95]. Phytoestrogens encompass a large collection of structurally different compounds with common cellular functions. They possess a low estrogenic activity, which makes them selective estrogen receptor modulators [96], with theoretically an antiproliferative property. Modulators of ER transcriptional activity were reported to bear beneficial effects on several health conditions [97]. For instance, regular consumption of soy, a rich source of phytoestrogen, has been associated with cardiometabolic marker improvements [96]. However, some evidence states that the beneficial effects of phytoestrogen vary, attributing the discrepancy to the gut microbiota profile [98]. This ambivalence may be exacerbated when taking into account gender differences, as it appears that phytoestrogen metabolism has differential signature in men and women [99]. In addition, the protective effect of phytoestrogen administration against breast cancer depends on a critical period of consumption. Phytoestrogen was reported to be beneficial when introduced early enough to counter-effect the steroidal estrogen; however, it is believed to be a tumor promoter when consumption is increased later in life [100].

### 5.2. Phytoestrogen Classes

BCs in phytoestrogens include stilbenes, coumestans, lignans, and isoflavones. The major stilbene is resveratrol, an ERs agonist, naturally produced as an antifungal agent in grape skin and peanuts. Among coumestans, coumestrol holds the highest estrogenic effect, and it is found mainly in soybeans, clover, Kala Chana, alfalfa sprouts, sunflower seeds, and spinach. Coumestrol inhibits the enzymatic activity of 17β-HSD and aromatase [101]. It also exerts antiestrogenic activity 30- to 100-fold higher than that of isoflavones by binding to ERα and ERβ.

### 5.3. Lignans

Lignans are a large group of polyphenols present in plants, with the highest concentration in flaxseed. Other good sources include sesame seeds, wheat, tea, and fruits (mainly berries). Lignan precursors are metabolized into biologically active enterolignans (enterodiol and enterolactone) by intestinal bacteria. Enterodiol and enterolactone have weak estrogenic activity and work through non-estrogenic mechanisms. Their biological activities include anticarcinogenic and estrogenic or anti-estrogenic properties [102]. However, Welshons et al. [103], who performed an in vitro study, reported otherwise, which may support the theory of the importance of the gut microbiota profile in converting phytoestrogens into xenoestrogens. The human cytochrome P450 enzymes (P450s), namely CYP 1A1 and CYP3A4, play a crucial role in the metabolism of xenobiotics. Flaxseed upregulates CYP1A1 enzymes to boost the anti-angiogenic and the onco-protective E2 metabolite [104], while dietary polyphenols modulate CYP3A4 expression and activity. Although the liver is a major site of CYP3A4 activity, some evidence suggests that the small intestine is an important site for the metabolism of polyphenolics compounds. The role of gut microbiota in the metabolic fate of polyphenolic compounds is surfacing, displaying the complex interactions between dietary polyphenols and CYP3A4 [105].

### 5.4. Isoflavones

Isoflavones are produced almost exclusively by the bean family Fabaceae. They are mostly found in soybeans and, to a lesser extent, in other legumes. The well-known isoflavones include daidzein, genistein, glycitein, and biochanin A (BCA). Isoflavones have antioxidant, anti-cancer, anti-microbial, and anti-inflammatory properties [106], although some reports described the risk of tumorigenesis with high consumption [107]. It is worth noting that consumption of fermented food high in isoflavones is more advantageous to health than non-fermented counterparts.

## 6. Effects of Phytoestrogen on the Gender-Based Brain

The properties of phytoestrogen on the dimorphic brain has been eloquently described in a review by Sumien et al. [108]. There is a differential effect of soy isoflavones on the brain of men and women, which suggests customization of soy intake may be a potential targeted therapy. A short-term consumption of a high-soy diet (100 mg/d) was associated with enhancement in both short- and long-term memory and improvement in cognitive functions [109]. Similar findings were reported in the Soy and Postmenopausal Health in Aging (SOPHIA) study of postmenopausal women who consumed 110 mg/d soy isoflavones [110]. However, another 12-week study with a maximum dose of 60 mg/d did not detect any improvement in cognitive functions in post-menopausal women [111]. Interestingly, other reports described that intake of 60 mg/d resulted in cognitive improvement in several categories related to frontal lobes [112,113]. Although soy isoflavones have shown beneficial effects on the female brain, there have been inconsistent results with the male’s brain [114]. One report stated that chronic consumption of 116 mg/d soy isoflavones improved spatial memory in men, but the impact was higher in women [115]. Based on an animal study, male rats required higher levels of estrogen to induce wheel running compared to their female counterparts [116]. This cross-link between laboratory and epidemiological observations suggests that lower levels of soy isoflavones may be needed to improve cognitive functions in women, which may be attributed to the higher expression of ERs in the female brain and the differential distribution [117].

## 7. Prominent BC in the Mediterranean Diet

Since the Mediterranean diet has been linked to several brain and mental health benefits, and it is hard to cover in detail every component, this section is dedicated to the concise description of BC-rich food, commonly consumed by the endogenous population, and their brain effects.

### 7.1. Omega-3 Fats

Marine fish are typically high in EPA and DHA, which both contribute to brain health. DHA promotes hippocampal neurogenesis and synaptic plasticity [118], in addition to a spectrum of physiological processes ranging from neurotransmitter release to intracellular signaling and axonal myelination [119]. DHA contributes to gray matter (GM) and subcortical volumes, mostly due to its embedment in membrane phospholipids [120]. Depletion models of DHA induce loss of neurites and reduce synaptic plasticity, which explains the disturbance in neurotransmission and reduction in gray matter volume [121] with the Western diet. On the other hand, EPA’s crucial role is maintenance of brain structure integrity. EPA mediates its neuroprotective roles through a cascade of cellular events that conclude with the production of anti-inflammatory molecules [122]. Both EPA and DHA bear anti-apoptotic, antioxidative stress, and pro-neurogenesis properties [123]. In addition, EPA supports progenitor neural cell proliferation, while DHA stimulates their differentiation into functional neurons [124]. Most of the anti-inflammatory and anti-apoptotic effects of omega-3 fats in the brain have been attributed to G protein-coupled receptor 120 (GPR120) activation [125]. However, neurogenesis and neuroplasticity are potentially modulated through NR PPAR-γ signaling [68]. Interestingly enough, it seems that there is a sex-related differential expression of PPARs in the brain [126], which suggests that a differential dose of EPA and DHA may be needed to optimize brain health in men and women. A recent direction in the field of personalized medicine is to account for genetic polymorphisms to optimize therapeutic outcomes. For instance, APOE4 allele carriers have impaired brain transport of non-esterified DHA into the brain; however, phosphatidylcholine-DHA is able to cross the inner membrane of the BBB, which reflects a potential therapeutic effect for Alzheimer’s disease patients [127].

### 7.2. Nuts

Nuts and seeds are essential components of the Mediterranean diet as they can be consumed as snacks and food garnish in many Mediterranean cuisines. Several epidemiological studies attributed health benefits to eating nuts, such as improvements in cardiovascular health, blood pressure, and blood sugar, among others. Nuts are a good source of fiber, monounsaturated fatty acids, and polyunsaturated fatty acids such as omega-6 and omega-3 fats, vitamin E, folate, magnesium, potassium, and calcium. Particularly, walnuts are an excellent source of the 18-carbon alpha-linolenic acid (ALA), an essential omega-3 fatty acid. However, because of the constrained conversion of ALA to EPA and DHA, especially with a high ratio of omega-6/omega-3 intake, nuts are not a considerable source of EPA. However, nuts derive most of their health benefits from fiber and their polyphenolic compounds, besides their micronutrient repertoire. The major polyphenol in walnuts is the hydrolyzable ellagitannin, pedunculagin, which is chemically modified after consumption and further processed by the gut microbiota to produce the different urolithin metabolites A, B, C, and D with a myriad of biological activities. Urolithin A is a mitophagy activator, and urolithin B carries antiproliferative and antioxidant activities, with evidence linking it to improved cognitive functions and attenuation of neuronal apoptosis in post-traumatic injury [128,129]. Urolithin C is a glucose-dependent activator of insulin secretion with an anti-tumor effect [130,131], while urolithin D’s function is still under investigation.

### 7.3. Olive Oil

Olive oil is obtained from the olive tree fruit with a botanical name Olea Europea. It is rich in polyphenols, oleic acid, squalene, and tocopherols (Vitamin E), which have been linked to improvement in metabolic functions and cardiovascular and brain health [132,133]. The phenolic repertoire of olive oil provides a spectrum of anti-inflammatory, antioxidant, anti-atherosclerotic, and anti-microbial properties that provide neuroprotective functions [134]. Although many of the health benefits of olive oil are typically credited to its high oleic acid concentration, oleuropein is the most reported BC with health benefits, and it is typically found in the olive skin (mostly in green olives) and their leaves [135,136]. Oleuropein, which is responsible for the bitter and pungent taste of extra virgin olive oil, has a high antioxidant capacity [136]. Olive oil, namely its least processed form or extra virgin olive oil (EVOO), has been associated with beneficial effects on brain functions conferring neuroprotection and neuroplasticity [137]. EVOO is richer in the phenolic extracts than the more processed olive oil, activating antioxidative and anti-inflammatory pathways, such as NRF2 signaling, and upregulating crucial anti-aging genes such as sirtuin-1 (SIRT1) [138]. EVOO and its bioactive compounds positively influence the gut microbiota profile, consequently improving intestinal integrity and supporting the growth of beneficial microorganisms. Most of these findings were extrapolated from animal models. However, these conclusions were indirectly demonstrated by the PREDIMED-NAVARRA randomized trial, which reported that long-term consumption of extra virgin oil significantly improved cognition in patients at high vascular risk [139].

### 7.4. Garlic and Onions

Both garlic (*Allium sativum* L.) and onion (*Allium cepa* L.) contain a spectrum of BCs such as organosulfur compounds, saponins, phenolic compounds, and polysaccharides [140,141,142]. The major active components in onions and garlic are sulfur-containing compounds, although they may differ in types and amounts.

Interestingly, garlic contains over 20 phenolic compounds, topping most vegetables [143]. The main phenolic compounds are β-resorcylic acid, pyrogallol, gallic acid, rutin, protocatechuic acid, and quercetin [141], which explains the long list of health benefits associated with regular garlic consumption. Red onion has the highest concentration of anthocyanins and flavonols, followed by yellow onion, and then white onion [144]. Quercetin is the major phytochemical from the flavonoid family in the skin of red onion, while quercetin-4-glucoside is the main compound in the bulb [145]. A comparative study suggests that garlic has higher antioxidant activity level than onions, mostly due to the lower organosulfur compounds content [145]. However, red onions contain more quercetin compared to white onions and garlic [140]. Both garlic and onions share common health benefits such as antioxidant, anti-microbial, anti-inflammatory, immune-modulatory, anti-hypertensive, anti-hyperlipidemic, and anti-cancer effects and hepato-, renal-, and neuro-protective characteristics [30,31,140,142,146,147,148,149].

### 7.5. Berries

Berries contain multiple phenolic compounds, such as anthocyanins, flavanols, proanthocyanidins, ellagitannins, and phenolic acids. Berries owe their dark rich colors to the pigment anthocyanins, which are robust antioxidants. Berry consumption has been linked to improvement in cognitive function, potentially due to enhanced blood flow and mental well-being and brain volume preservation [150,151]. It is possible that polyphenols strengthen neuroplasticity and neurogenesis, which eventually leads to these neural enhancements. Blueberries are the most studied among the berry family, and they are often referred to as “superfoods” because of their long list of health benefits. Some of the described brain benefits include attenuation in microglial activity, modulated nitric oxide production, and enhancement in neuroplasticity [150,152].

### 7.6. Pomegranate

Pomegranate, from the Punica granatum species, is rich in ellagitannins and gallotannins, anthocyanins and procyanidins, catechins, epicatechin, and quercetin, which have been reported to enhance several brain functions, such as a decrease in oxidative/inflammatory stress signaling and an increase in antioxidant protective effects [94,153]. Ellagitannins and gallotannins are hydrolysable tannins that tend to be the most prevalent in nuts and berries. Pomegranate possesses a strong antioxidant activity, thanks to its consortium of BCs with strong bioactivity, and anthocyanins are of high biological significance among this fruit’s phenolic compounds.

### 7.7. Green Tea

The degree of tea leaf processing and oxidation produces six different types: green, white, yellow, oolong, black, and dark tea. Although green and black may be consumed in Mediterranean countries [32], tea has been known to be a healthy commodity. The degree of tea leaf oxidation impacts the level of its antioxidant capacity. Green tea is non-oxidized, which explains why it has higher antioxidant potency compared to oxidized teas like oolong (semi-oxidized) and black tea (completely oxidized). Green tea is known to bear the highest health benefits as it is a good source of polyphenolic compounds, particularly epigallocatechin gallate (EGCG), catechin, galactatechin, epigallocatechin, epicatechin, and galactoatechin gallate. These BCs were mostly described in vitro and in vivo studies as potent antioxidants [154].

### 7.8. Coffee

Although coffee drinking is a universal phenomenon, coffee is among the most highly consumed beverages in the Mediterranean diet. Several reports in the literature linked low–moderate consumption of coffee (as regular or decaffeinated) to improvement in several health parameters and brain functions, such as cognitive and mental well-being. The coffee bean is rich in chlorogenic acid, a family of esters [155]. It also contains methyl-xanthine alkaloid caffeine and melanoidins [156]. Although caffeine is a nervous system stimulant, the described improvement in brain functions was also attributed to the BC in the coffee bean [157]. Pharmacodynamic and pharmacokinetic polymorphisms of CYP1A2 and the ADORA2A genes explain individual responses in caffeine metabolism and its clearance from the body, impacting several brain-related outcomes such as sleep, anxiety, and cognitive functions [158]. There are gender differences in caffeine kinetic metabolism, which also warrants further research in combination with genetic polymorphisms.

## 8. Strengths and Limitations

This review centralized several pieces of evidence to bring to light the need for better appreciation of the therapeutic value of BCs by the healthcare community while proposing gaps in the literature to support further research. The topic of polyphenols in relation to brain health is very broad and complex. This review only highlighted a major part of the polyphenol realm. Future work should address the different phytochemicals not covered in this review and the complex interaction with the microbiome in relation to brain functions. Another interesting topic that this review breezed over is crucial signaling pathways, such as the nrf2/ARE pathway, in relation to phytochemicals and brain functions.

## 9. Bridging the Gap between Science and Practice

The evidence compiled in this review clearly implies that a diet high in BCs may provide a prophylactic and therapeutic effect against several brain ailments. There is a need to expand our understanding of other ethnic diets on brain health, such as the South American, Asian, African, and Middle Eastern diets. The need to move in this direction is becoming urgent as ultra-processed food is topping food consumption and, therefore, impacting gut and brain health. Although BCs have demonstrated beneficial aptitudes, fine-tuning types and dosage based on biological sex and brain age may present prophylactic and therapeutic potential similar in strength to pharmaceutical agents, with presumably minimal side effects. With the surged interest in personalized medicine, customized diets may become attractive not only to nutritionists but also to a list of practitioners working in consortiums for patients’ care. Currently, healthcare therapies follow the “one-size-fits-all” approach. Personalizing therapies promote an effective healthcare outcome and reduces its cost. Historically, therapies have been developed based on human physiological and behavioral research results. With the advancements in molecular nutrition, epigenetics, and the microbiome, new layers of complexities are merging. There is a need to home in on customizing therapies and move from “let thy food be thy medicine, and thy medicine be thy food” into “let thy food be thy personalized medicine, and thy personalized medicine be thy food”.

## 10. Conclusions

Research on bioactive compounds suggests robust prophylactic and therapeutic roles in fighting a spectrum of ailments, particularly the ones related to the nervous system. Their anti-inflammatory and antioxidative effects, combined with their gene expression and microbiota modulatory actions, make them an attractive remedy without the side effects of pharmaceuticals. However, we are far from prescribing dietary regimens rich in BCs for therapeutic purposes as evidence-based guidelines are still lacking, in part due to the underrated remedial effect of nutrient-dense diets. Therefore, as the interest in personalized medicine increases, focused research and funding should be the cornerstone to reach that goal.

## Data Availability

Not applicable.

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
