# Peer review of "Bioactive Compounds for Customized Brain Health: What Are We and Where Should We Be Heading?"

_ijerph, 2023, doi:10.3390/ijerph20156518_

Round 1

Reviewer 1 Report (Previous Reviewer 1)

Thank you for making the appropriate edits.  It is an intense paper--thank you for supporting with more detail.  I only found now three spelling issues but that is the extent:

Line 44: “Slowing emerging” should be “Slowly emerging”

Line 146 – Dysfunction misspelled

Line 570 – is there an error on this line?  , and an increase in alzheiprotective effects—did you mean to spell out Alzheimers

Line 626-- toping should be topping

Author Response

Thank you for providing us with constructive feedback. Please find point by point responses.

Reviewer 1

Thank you for making the appropriate edits.  It is an intense paper--thank you for supporting with more detail.  I only found now three spelling issues but that is the extent:

Line 44: “Slowing emerging” should be “Slowly emerging”

Response: Thank you for noting it. It has been fixed

Line 146 – Dysfunction misspelled

Response: Thank you for noting. It has been fixed

Line 570 – is there an error on this line?  , and an increase in alzheiprotective effects—did you mean to spell out Alzheimers

Response: Thank you for noting it. It has been fixed

Line 626-- toping should be topping

Response: Thank you for noting it. It has been fixed

Reviewer 2 Report (New Reviewer)

The current review article had two major purposes. The first purpose was to explore the prominent bioactive compounds (BCs) that influence cognitive and mental health to deliver a potentially prophylactic framework for practitioners. The second purpose was to identify gaps in the literature in an effort to suggest new research agendas for scientists.

 The review was extensive and covered a number of interrelated topics relative to these general purposes. Most of these topics are being extensively researched at the current time and there is great interest in many of these. The review consisted of some general nutrition background information and background about the brain functions that are influenced by nutrition. In addition, an array of topics such as sex hormones and gender differences, BCs, neuroprotection, gut microbiota, polyphenols, phytoestrogens, lignans, isoflavones, and the Mediterranean diet and its components are all covered. Finally, some suggestions for future research and conclusions were covered.

 Overall, I think this review was relatively well-written, covers important topics, and provided a great deal of interesting information on the above topics. Therefore, the overall content of the article was pretty good. I think it will be of interest to readers of the journal and adds to the literature on the topic. Thus, I think the paper should eventually be published after revisions are made. Most of the revisions are minor, but have to be made before the review is publishable.

Comments:

Overall, the article was well-written in a global sense and laid out in a reasonable order and framework. However, the main issue with the paper are smaller aspects of the writing and formatting. There are numerous instances of various types of typos, wording mistakes, awkward wording, grammatical errors, bibliographical errors etc. that need attention and proofreading. There are far too many to name individually so I will give just some examples below mainly for the abstract and introduction, but also some of the more major ones in other parts of the paper.

1.      Abstract:

a.       Line nine “but” should read “but also”

b.      Line 11, “produced” should read “have produced”

c.       “dig” use a different word

d.      Line 18, “regarding” use a different wording such as “prominent BCs that can impact cognitive function”

e.       Lines 19 and 20, “provide” is a poor choice of words, “identify gaps” for instance would be much better, you are not providing or giving a gap, you are pointing them out afterall.

2.      Introduction:

a.       well recognized should read “well-recognized”

b.      Line 28, more spaces after the period at the end of the sentence than other sentences. This type of error is present numerous times throughout the paper.

c.       Line 36 vs Line 38. “Bioactive” and “bio-active” one is hyphenated one is not.

d.      Line 49, “accurate conclusion” should read “accurate conclusions”

 3.      Various selected other errors or clarifications.

a.       Lines 87 and 88 should read “the BBB”

b.      Line 88 “anatomical damage” not “anatomical damages”

c.       Line 100 and throughout the paper. Is “gender brain” correct terminology. Should it be “gender-based differences” or “biological sex differences” or something like that instead?

d.      Line 114 “women low mood” is not correct, not sure what is trying to be said “women with low mood”?

e.       Line 135 dysfunction is misspelled.

f.        Line 139 “were” should read “have been”

g.      Line 152 should read “rather than therapeutically

h.      Line 178, should “plant-forward” read “plant-based”? I could be wrong here.

i.        Line 186, “is” should read “are”

4.      Bibliography: There are many mistakes here. I just will illustrate a few instances below of the major problems. Check all bibliography formatting.

a.       The titles of many journals are not written consistently. Some are abbreviated, some are not, some capitalize all the letter of the journal title some do not. For example look at the differences between references 1, 6 and 8 versus most of the other journal titles where all words are capitalized.

b.      For instance references 22 and 24 use abbreviations for the journal title where most of the other references do not for instance 26 and 34 etc etc.

c.       43 has the title of the article every single letter in all caps, no other references have that.

See my comments to the authors.

Author Response

Your response to the comments:

Thank you to all reviewers who provided us with constructive feedback. Please find point by point responses.

Reviewer 2

The current review article had two major purposes. The first purpose was to explore the prominent bioactive compounds (BCs) that influence cognitive and mental health to deliver a potentially prophylactic framework for practitioners. The second purpose was to identify gaps in the literature in an effort to suggest new research agendas for scientists.

The review was extensive and covered a number of interrelated topics relative to these general purposes. Most of these topics are being extensively researched at the current time and there is great interest in many of these. The review consisted of some general nutrition background information and background about the brain functions that are influenced by nutrition. In addition, an array of topics such as sex hormones and gender differences, BCs, neuroprotection, gut microbiota, polyphenols, phytoestrogens, lignans, isoflavones, and the Mediterranean diet and its components are all covered. Finally, some suggestions for future research and conclusions were covered.

Overall, I think this review was relatively well-written, covers important topics, and provided a great deal of interesting information on the above topics. Therefore, the overall content of the article was pretty good. I think it will be of interest to readers of the journal and adds to the literature on the topic. Thus, I think the paper should eventually be published after revisions are made. Most of the revisions are minor, but have to be made before the review is publishable.

Comments:

Overall, the article was well-written in a global sense and laid out in a reasonable order and framework. However, the main issue with the paper are smaller aspects of the writing and formatting. There are numerous instances of various types of typos, wording mistakes, awkward wording, grammatical errors, bibliographical errors etc. that need attention and proofreading. There are far too many to name individually so I will give just some examples below mainly for the abstract and introduction, but also some of the more major ones in other parts of the paper.

  1. Abstract:
  2. Line nine “but” should read “but also”

Response: thank you for noting it. It was fixed

  1. Line 11, “produced” should read “have produced”

Response: thank you for noting it. It was fixed

  1. “dig” use a different word

Response: It was changed to dive

  1. Line 18, “regarding” use a different wording such as “prominent BCs that can impact cognitive function”

Response: thank you for the comment. We changed it accordingly

  1. Lines 19 and 20, “provide” is a poor choice of words, “identify gaps” for instance would be much better, you are not providing or giving a gap, you are pointing them out afterall.

Response: thank you for the comment. We changed it accordingly

  1. Introduction:
  2. well recognized should read “well-recognized”

Response: Done

  1. Line 28, more spaces after the period at the end of the sentence than other sentences. This type of error is present numerous times throughout the paper.

Response: Thank you for this comment. We fixed it throughout the manuscript.

  1. Line 36 vs Line 38. “Bioactive” and “bio-active” one is hyphenated one is not.

Response: Done

  1. Line 49, “accurate conclusion” should read “accurate conclusions”

Response: Done

Various selected other errors or clarifications.

Response: The manuscript has been revised

  1. Lines 87 and 88 should read “the BBB”
  2. Response: Done
  3. Line 88 “anatomical damage” not “anatomical damages”
  4. Response: Thank you for this comment . Done
  5. Line 100 and throughout the paper. Is “gender brain” correct terminology. Should it be “gender-based differences” or “biological sex differences” or something like that instead?
  6. Response: Thank you for this comment . Done
  7. Line 114 “women low mood” is not correct, not sure what is trying to be said “women with low mood”?

Response: Thank you for this comment. Low was replaced with ‘depressed’

  1. Line 135 dysfunction is misspelled.

Response: Thank you noting it. It was fixed

  1. Line 139 “were” should read “have been”

Response: Thank you noting it. It was fixed

  1. Line 152 should read “rather than therapeutically

Response: Thank you noting it. It was fixed

  1. Line 178, should “plant-forward” read “plant-based”? I could be wrong here.

Response: this was a suggestion from a reviewer use forward instead of based.

  1. Line 186, “is” should read “are”

Response: Thank you noting it. It was fixed

  1. Bibliography: There are many mistakes here. I just will illustrate a few instances below of the major problems. Check all bibliography formatting.
  1. The titles of many journals are not written consistently. Some are abbreviated, some are not, some capitalize all the letter of the journal title some do not. For example look at the differences between references 1, 6 and 8 versus most of the other journal titles where all words are capitalized.
  2. For instance references 22 and 24 use abbreviations for the journal title where most of the other references do not for instance 26 and 34 etc etc.
  3. 43 has the title of the article every single letter in all caps, no other references have that.

Response 1-3 ( Bibliography): Thank you noting these issues. We reran the reference manager which generated the format based on the journal’s setting. We revised as much as possible the noted differences. Some journals are officially using the acronyms such as BMJ and FASB.

Reviewer 3 Report (New Reviewer)

In this manuscript, the main bioactive components related to cognitive functions and mental health have been compiled. In addition, the brain morphology of male and female genders, the relationship between hormonal order and nutrition and health outcomes due to gender differences, and health outcomes inclusive human behavior are explained in this study and the manuscript is sufficiently expanded in the present form.

Some corections are needed:

1-The font color is not black all over the article text.

2-Attached is the plagiarism report. Please correct manuscript.

Author Response

Thank you for providing us with constructive feedback. Please find point by point responses.

Reviewer 3

In this manuscript, the main bioactive components related to cognitive functions and mental health have been compiled. In addition, the brain morphology of male and female genders, the relationship between hormonal order and nutrition and health outcomes due to gender differences, and health outcomes inclusive human behavior are explained in this study and the manuscript is sufficiently expanded in the present form.

Response: Thank you for your comments

Some corections are needed:

1-The font color is not black all over the article text.

Response: We double checked the font from our end, it was black. We reapplied the black color.

2-Attached is the plagiarism report. Please correct the manuscript.

Response: Thank you for providing the report. Most of the highlighted items are technical words that we cannot change and or best fit words. However, we did our best to improve the score. Hope the reviewer is satisfied with the edits.

Reviewer 4 Report (New Reviewer)

In line 135, it is stated that "Menopausal women and men have higher nutrient needs for choline," which is a mistake since men do not go through menopause. Although menopause and andropause have some similarities, they are different processes with specific characteristics.

In line 165, it would be important to mention the main differences observed in the effects based on gender.

In line 257, it would be beneficial to provide a more elaborate explanation of how serotonin is managed by the microbiota. Is there a mechanism described on how this happens?

In line 285, it would be valuable to mention some of the results obtained in those few reports to highlight the importance of this symbiotic relationship.

In line 300, two classes of polyphenols are mentioned, "flavonoids and non-flavonoids," and the flavonoids are described in terms of their characteristics and sources. However, non-flavonoids are not described in the same way, so it would be worth including a description of them as well.

The information in line 315 is repeated in line 317, which can be addressed by consolidating or rephrasing the content to avoid redundancy.

Regarding the data on "bioavailability and cellular mechanisms," a more organized presentation focusing on the characteristics required to cross the blood-brain barrier (BBB) would be helpful. Additionally, it would be convenient to mention the LogP values of those polyphenols that have been reported capable of crossing the BBB.

Minor editing of English language required

Author Response

Thank you for providing us with constructive feedback. Please find point by point responses.

Reviewer 4:

In line 135, it is stated that "Menopausal women and men have higher nutrient needs for choline," which is a mistake since men do not go through menopause. Although menopause and andropause have some similarities, they are different processes with specific characteristics.

Response: Thank you for pointing this out. The text meant menopausal women and men as all men. We reworded the sentence to reflect this aim.

In line 165, it would be important to mention the main differences observed in the effects based on gender.

 Response: Thank you for this comment. We added a statement to cover this request.

In line 257, it would be beneficial to provide a more elaborate explanation of how serotonin is managed by the microbiota. Is there a mechanism described on how this happens?

 Response: Thank you for idea. We have added a statement to further explain the mechanism.

In line 285, it would be valuable to mention some of the results obtained in those few reports to highlight the importance of this symbiotic relationship.

 Response: we have included further information

In line 300, two classes of polyphenols are mentioned, "flavonoids and non-flavonoids," and the flavonoids are described in terms of their characteristics and sources. However, non-flavonoids are not described in the same way, so it would be worth including a description of them as well.

 Response: Thank you for observation. We included further information.

The information in line 315 is repeated in line 317, which can be addressed by consolidating or rephrasing the content to avoid redundancy.

 Response: Thank you for this observation. We deleted the redundancy.

Regarding the data on "bioavailability and cellular mechanisms," a more organized presentation focusing on the characteristics required to cross the blood-brain barrier (BBB) would be helpful. Additionally, it would be convenient to mention the LogP values of those polyphenols that have been reported capable of crossing the BBB.

Response: Thank you for this comment. We included further information on the molecular weight of the polyphenols that cross BBB.

This manuscript is a resubmission of an earlier submission. The following is a list of the peer review reports and author responses from that submission.

Round 1

Reviewer 1 Report

See attached
